# MRN (MRE11-RAD50-NBS1) Complex in Human Cancer and Prognostic Implications in Colorectal Cancer

**DOI:** 10.3390/ijms20040816

**Published:** 2019-02-14

**Authors:** Yiling Situ, Liping Chung, Cheok Soon Lee, Vincent Ho

**Affiliations:** 1School of Medicine, Western Sydney University, Penrith, NSW 2751, Australia; ysitu.dr@gmail.com (Y.S.); liping.chung@westernsydney.edu.au (L.C.); soon.lee@westernsydney.edu.au (C.S.L.); 2Ingham Institute for Applied Medical Research, Liverpool, NSW 2170, Australia; 3Department of Anatomical Pathology, Liverpool Hospital, Liverpool, NSW 2170, Australia; 4Discipline of Pathology, School of Medicine, Western Sydney University, Campbelltown, NSW 2560, Australia; 5Faculty of Medicine, South Western Sydney Clinical School, University of New South Wales, Liverpool, NSW 2170, Australia; 6Faculty of Medicine and Health, Central Clinical School, University of Sydney, Camperdown, NSW 2050, Australia

**Keywords:** DNA damage response, MRE11-RAD50-NBS1 complex, colorectal cancer, biomarkers, prognosis, radiosensitivity

## Abstract

The MRE11-RAD50-NBS1 (MRN) complex has been studied in multiple cancers. The identification of MRN complex mutations in mismatch repair (MMR)-defective cancers has sparked interest in its role in colorectal cancer (CRC). To date, there is evidence indicating a relationship of MRN expression with reduced progression-free survival, although the significance of the MRN complex in the clinical setting remains controversial. In this review, we present an overview of the function of the MRN complex, its role in cancer progression, and current evidence in colorectal cancer. The evidence indicates that the MRN complex has potential utilisation as a biomarker and as a putative treatment target to improve outcomes of colorectal cancer.

## 1. Introduction

Colorectal cancer (CRC) is the second most commonly diagnosed malignancy in females, the third most commonly diagnosed malignancy in males, and has the third highest global mortality rate [1,2]. The high incidence of colorectal cancer may be attributed to the Western lifestyle; factors that increase risk include obesity, unhealthy dietary habits, and smoking [3]. Colorectal cancer develops from premalignant polypoid lesions, and thus the introduction of better screening programs, early detection, and prompt removal of premalignant lesions via endoscopy have assisted in the reduction in incidence by 2% to 3% per annum [4,5]. This rate does not, however, reflect rectal cancer as an isolated entity. Colonic, rectal, and anal cancers are often combined as a single entity although they differ in their biological behaviour, metastatic pattern, clinical treatment methods, and relapse rate [6]. 

Surgical resection combined with chemotherapy is the standard treatment for colon cancer, whereas surgical resection combined with neoadjuvant radiotherapy is preferred for rectal cancer [7,8]. Surgical resection for rectal cancer is usually insufficient due to limited access resulting from the anatomical location of the rectum amongst the mesorectal fascia, and other pelvic organs [9]. A clear tumour margin can be difficult to ascertain, and thus, neoadjuvant radiotherapy with or without chemotherapy is often used to achieve tumour regression and clinical downstaging [10]. The use of neoadjuvant radiotherapy has improved overall survival and reduced tumour recurrence. Despite this, the overall survival of rectal cancer patients remains poor as a consequence of both tumour recurrence and complications of chemoradiation. Thus, individualised treatment methods are important, and can be assisted by new imaging modalities, improved surgical techniques, and better understanding of tumour behaviour and response to chemoradiation [11]. 

The aim of radiotherapy and chemotherapy is to create excessive DNA damage in cancer cells, which then results in apoptosis. However, there is substantial collateral damage to nearby healthy cells, which results in disabling side-effects. Understanding the underlying molecular behaviour of tumour cells will allow targeted therapy and individualised treatment to improve outcomes and reduce unwanted side-effects. 

The DNA damage response (DDR) pathway is a system whereby intra- and intercellular interactions allow for detection of DNA damage, halting cellular replication, repairing DNA damage, and re-establishing a healthy state [12]. More than 450 proteins have been identified as part of this system, and some of these have been studied in depth as potential targets for therapy and as biomarkers for predicting response to therapy [9,13]. The DNA damage response is primarily based on identification of double-strand breaks (DSBs). These breaks are repaired through two different systems: homologous recombination (HR) and non-homologous end joining (NHEJ). Homologous recombination is activated after DNA replication, using the undamaged sister chromatid as a repair template, and occurs primarily in the late S/G2 phase [14,15]. This is a relatively accurate and efficient process. In contrast, the non-homologous end-joining pathways are not reliant on replicated DNA, thus, more prone to error. The NHEJ pathways are often seen in the G1 phase prior to DNA replication, and this process may occasionally introduce DNA rearrangements and mutations [12]. As discussed below, the MRN complex is a key component of both HR and NHEJ repair systems of the DDR pathway. Here, we discuss the function of the MRN complex in the DNA damage response, its relationship to colorectal cancer, and the potential role of this protein complex in clinical treatment of colorectal cancer.

## 2. Structure and Function of the MRE11-RAD50-NBS1 (MRN) Protein Complex

The MRE11-RAD50-NBS1 (MRN) complex plays an important role in DNA damage response (DDR) and the repair pathway of double-strand breaks (DSBs). Human DNA is under constant insult from exogenous and endogenous factors that lead to genomic instability. The cell employs specific molecular systems to detect and signal the presence of DNA damage, induce cell cycle arrest, and repair damaged DNA or induce apoptosis if repair is unsuccessful. The accumulation of DNA damage over time is a hallmark of cancer. 

The MRN complex is heterotrimer consisting of meiotic recombination 11 (MRE11), DNA repair protein Rad50 (RAD50), and Nijmegen breakage syndrome 1 (NBS1) (Figure 1A). It is the major catalytic protein complex in coordinating and sensing DSBs and in initiating the DNA damage response pathway. Once the DDR pathway is activated, it results in activation of ataxia telangiectasia mutated (ATM) dimers, repairing DSBs through complex machinery at different stages including cell cycle checkpoints, telomere length maintenance, and meiosis [15,16,17]. This is done via extensive protein-protein interactions, ATP hydrolysis, endonuclease and exonuclease activities, and coordinating re-joining of repaired DNA strands [18]. Extensive studies of the MRN complex in-vitro have elucidated the dynamics and complexity of the protein, however, there are no standardized assays or interpretation of the MRN complex currently available for research use. Nonetheless, there is consistency between the MRN complex and its interactions and activation of downstream molecules including ATM dimers, γ-H2AX, chk1, chk2, and p53 [19]. 

The initial DSBs are detected by the MRN complex (Figure 1B). The MRN complex is recruited to foci of DNA damage through interactions with phosphorylated H2A histone family member X (γ-H2AX), particularly the NBS1 subunit [20,21,22]. Subsequently, MRN complex recruits the inactive ATM dimer [23]. The ATM protein kinase plays a pivotal role in the downstream coordination of the DDR pathway, including check-point kinase 1 and 2 (chk1, chk2), BRCA1, and p53 [24]. It also functions downstream, where it phosphorylates the three MRN subunits MRE11, RAD50, and NBS1 [15]. The activated ATM dimer interacts with chk2 to further inactivate chk1, which leads to cell cycle arrest in G2/M [19]. This allows for DNA damage repair to occur prior to cellular replication [17]. Phosphorylated NBS1 provides the binding platform for MRN‒ATM complexes [20,25]. In the homologous recombination repair pathway, MRE11 exonuclease activity excise and sculpts the DNA from the break point to about 200 base pairs downstream of the DSB [25]. CtIP and RAD51 also assist in processing the DNA ends to create single-stranded DNA (ssDNA) 3′ overhangs [20,26,27]. RAD50 plays a significant role in joining these ssDNA overhangs [25] and in telomere length maintenance [28]. BRCA2 regulates RAD51 recombinase, and initiates ssDNA pairing during homologous recombination (HR) [29]. Additionally, the MRE11 subunit assists in non-homologous end joining (NHEJ), which processes and directly joins DNA ends in a process involving other molecules such as Ku70/80 at the G2/M checkpoint [30]. Through utilising the MRN complex and a combination of other molecular mechanisms, DSBs can be repaired promptly, restoring genomic stability prior to cellular division. 

## 3. The Role of the MRN Complex in Colorectal Cancer 

It is known that particular homozygous mutations of the NBS1 gene lead to Nijmegen breakage syndrome, and increase the risk of haematological cancers [31], while defective mutations in MRE11 lead to the development of ataxia-telangiectasia-like disorders. Nonetheless, the role of the MRN complex and its components in tumorigenesis remains controversial. Since the MRN complex is essential for DSB repair and telomere maintenance, the MRN complex and its components have been studied in multiple cancer lines including colorectal cancer, nasopharyngeal cancer, gastric cancer, prostate cancer, breast cancer, low-grade ovarian cancer, endometrial cancer, lung cancer, and Lynch syndrome [32,33,34,35]. It is postulated that defective function and low expression of the MRN and its components leads to an increase propensity for cellular destabilisation, accumulation of DNA damage, and malignant transformation. However, also owing to its ability to repair DNA DSBs, it is likely that the levels of MRN expression influence the response of cancer cells to chemotherapy, radiotherapy, and the degree of apoptosis. Conflicting data obtained in different cell lineages suggest a complex relationship between MRN expression and carcinogenesis. While some studies have associated high expression of MRN and its components with poorer outcome and treatment resistance to chemoradiation [18,34,36,37], others have found the opposite.

Because of the importance of the MRN complex in the DNA damage response cascade, the MRN complex and its components remain promising as biomarkers for predicting treatment response and as targets for cancer therapy. Although there are differing opinions about the relationship between MRN expression and clinical outcomes in different forms of cancer, there is evidence that the MRN complex is of significance in colorectal cancer (see Table 1 and Table 2). 

Early studies of the MRN complex have identified MRE11 as a core component, which plays an important role in the stability of RAD50 and NBS1. Mutations in MRE11 are associated with increased susceptibility to colorectal cancer, especially in relation to increased microsatellite instability [41,44]. MRE11 mutations occur in the majority of mismatch repair (MMR)-defective primary colorectal cancers, and approximately 15% of colorectal cancers display microsatellite instability (MSI) [14,45]. The MMR system plays a crucial role in identifying errors in microsatellites via activities of one or more of its component genes MLH1, MSH2, MSH6, and PMS2. Defects in function of any of these genes leads to accumulation of DNA damage and MSI [14]. 

One of the earlier studies focused on homozygous mutations of the poly(T)11 repeat within the human MRE11 intron 4. This mutation is found exclusively in MMR-deficient primary colorectal tumours, and leads to splicing errors, reduced MRE11 expression, and truncated MRN protein complex [46]. It is suspected that MRE11 deficiency causes microsatellite instability through defective interactions with MLH1 and MRE11 leading to their inactivation in MMR-deficient cancers [41,44]. Several studies have confirmed defective MRE11 expression in MSI colorectal cancers, and interestingly, a mild defect in either MRE11 or RAD50 is adequate to downregulate RAD50 and overall MRN expression in MSI tumours [47]. Further studies have aimed to reproduce these findings by correlating the degree of MRN expression with clinical outcomes. Alemayehu et al. [33] assessed MRN expression and its relationship with the degree of MSI instability and extent of alterations. The study confirmed a high degree of mutation in MRE11, RAD50, or both genes in the majority of Lynch-syndrome, but failed to demonstrate a clear relationship between destabilisation of MRN complex and higher degrees of MSI. A possible explanation for this finding is that genetic variations and mutations in MRE11 and RAD50 do not correlate with their expression, and thus have not demonstrated clinical relevance [48]. It is postulated that low expression of the MRN complex reduces the function of MLH1 leading to increased MSI, DSB repair deficiency, and genomic instability [44]. Another study postulated that reductions in MRE11 expression will increase chemotherapy sensitivity due to the impaired repair and signalling of DSBs [40]. This study demonstrated a relationship between MRE11-positivity and undifferentiated types of colorectal cancer, and showed less reduction in tumour size and poorer progression-free survival compared to MRE11-negativity in response to chemotherapy; however, these results were not clinically significant. 

Microsatellite stable (MSS) colorectal tumours, however, behave differently, and there is equal, if not increased expression of intact MRN complex in microsatellite stable CRCs compared to surrounding tissue [43,49]. Gao et al. observed a relationship among MSS tumours between high MRN expression and earlier tumour grade and favourable survival. Increased MRE11 also increased tumour cell apoptosis [43]. This suggests that the MRN complex may play different roles in different colorectal cancer subtypes, and in particular, adding to the tumorigenesis of MSI CRCs. Thus, in the subgroup of patients with MSI CRCs, any germline mutations in the MRN complex may produce additional genomic instability, and therefore may provide a target for treatment.

Because of the close relationship between MRN complex and colorectal cancer, it has been proposed that the MRN complex may provide possible biomarkers for predicting disease progression and treatment response. In a study by Chen et al. [38], upregulation of RAD50 was found in colorectal cancer cells and radiotherapy-resistant colorectal cancer cells. Elevated RAD50 was associated with poor patient survival, and experimentally knocking out RAD50 increased sensitivity to irradiation in colorectal cancer. This supports the notion that RAD50 is crucial for the success of repairing DSBs, and the downregulation of this impairs the function of MRN, thus leading to higher degree of apoptosis within irradiated cancerous cells. Another MRN subunit, NBS1, plays a key role in ATM phosphorylation and in the formation of the MRN‒ATM complex to initiate downstream binding of BRCA1 [42]. A study of Japanese subjects identified a naturally occurring germ line mutation of NBS1 (NBS1 IVS11+2insT) that leads to a non-functional carboxy terminus, which causes severe dysfunction in MRE11 and ATM binding. The C-terminus of NBS1 is required for recruitment and enhancement of ATM kinase activity [27]. Without a functional MRN‒ATM complex, downstream regulation of DDR is impaired, resulting in significant susceptibility to colorectal cancer. The presence of this mutation was associated with an increased odds ratio of 12.6 for gastrointestinal cancers, and 9.43 for colorectal cancer specifically [42]. These studies suggest that high RAD50 or low NBS1 is associated with increased tumour grade and treatment resistance. RAD50 expression is known to be associated with NBS1 expression [36], supporting the similar findings of these studies. 

Contrary findings were reported by Ho et al. [9], who demonstrated a relationship between low RAD50 expression and poorer disease-free survival and perineural invasion in rectal cancer patients (see Table 2). Low RAD50 expression in the tumour periphery was also observed to be increased in patients needing adjuvant therapy. However, there was no significant difference in tumour stage or lymph node metastasis associated with low RAD50. Microsatellite stability was not ascertained for specimens in this study, which may explain the disparate results, as Gao and colleagues [46] found that clinical outcomes associated with low RAD50 expression were specific for microsatellite stable colorectal cancers. 

Another study focusing on the MRN subunits MRE11 and ATM demonstrated a significant relationship between overexpression of these two proteins with poor disease-free survival in rectal cancer patients [39]. Overexpression of both MRE11 and ATM was also related to lymph node positivity, suggesting that the observed poor overall survival was likely due to aggressiveness and metastatic properties of the tumour cells. It is reasonable to attribute this to the increased levels of MRE11, which stabilises RAD50 and NBS1, and recruits ATM [17]. In isolation, however, MRE11 was not associated with any clinicopathologic characteristics or progression-free survival [39]. Similar findings were observed in another study of rectal cancer cells, in which elevated expression of MRN was associated with poor disease-free survival and overall survival in rectal cancer patients with and without neoadjuvant radiotherapy [37]. 

Ihara et al. investigated the response to treatment and prognosis of colorectal cancer patients treated with oxaliplatin, which is a DNA intra-strand cross-linking agent that induces DSBs in DNA [40]. MRE11 expression was not associated with any clinicopathologic characteristics, but there appeared to be better progression-free survival in MRE11-negative patients compared to MRE11-positive patients treated with oxaliplatin, although the results failed to reach statistical significance.

These findings indicate that the versatility of the MRN complex in response to exogenous insults is a consequence of complex interactions among its components. Although the evidence suggests that high MRN expression seems to increase treatment resistance, in the clinical setting, none of the individual MRN components have an advantage over the others or are individually associated with a particular clinical feature. We might speculate that MRE11 and RAD50 interact and autoregulate the expression and activation of one another, and the inconsistencies between studies can be attributed to the variability and versatility of these protein interactions. 

## 4. MRN Protein Complex and Sensitivity to Chemoradiation in Colorectal Cancer 

As demonstrated in multiple studies, components of the MRN complex are associated with treatment resistance and poorer prognosis in colorectal cancer [9,37,39]. This association supports the use of MRN components as biomarkers for predicting treatment response in colorectal cancer. The levels of ATM and γ-H2AX have been identified as biomarkers for radiosensitivity in various tumour cells [50]. Since MRN is closely related to the regulation of ATM and γ-H2AX, it too could be considered as a biomarker for colorectal cancer, especially in certain subtypes such as MMR-deficient CRCs. 

Furthermore, there appears to be a relationship between MRN expression and poor radiotherapy response. This is due to the underlying mechanism of ionizing radiation and the functions of the MRN complex. The aim of radiotherapy is to utilise ionizing radiation to induce double-strand breaks in DNA, taking advantage of the fact that an accumulation of DNA damage that exceeds the capacity of the DDR pathway to repair the damage will cause cell cycle arrest and apoptosis. Thus, colorectal tumours that are amenable to radiotherapy, but which also overexpress the MRN complex, will have added DNA repair efficiencies that contribute to treatment resistance. Ho et al. demonstrated that high expression of MRN complex proteins was associated with poor disease-free and overall survival; the correlation was particularly significant in the subgroup of patients receiving neoadjuvant radiotherapy [37]. Similarly, overexpression of MRE11 and ATM was found in rectal tumours after neoadjuvant radiotherapy, and was associated with poor overall survival [39]. Downregulation of RAD50 in isolation increases sensitivity of CRCs to irradiation [38]. Sheridan et al. [51] examined MRE11 expression, and showed that increased expression was found in the tumour periphery outside the neoadjuvant irradiated field. Peripheral MRE11 expression was higher than in the tumour core and in non-cancerous patients. However, there was no correlation between MRE11 and survival or radiosensitivity. Although not confirmatory, the notion remains that elevated MRN expression reduces radiosensitivity while a reduction in MRN expression increases tumour radiosensitivity, improving the success of tumour downgrading in neoadjuvant chemoradiation. Preclinical trials of ATR inhibitors have demonstrated anti-tumour effects in cancer cells with low MRE11 expression [52]. Since tumour cells rely heavily on the DDR pathway and the MRN complex, inhibiting the function of ATR inhibits the growth of colorectal cancers in vitro and has the potential to increase chemoradiation sensitivity [53]. 

Evidence supports the use of the MRN complex and its components as markers for predicting tumour resistance to chemotherapy. While the combination of MRE11 and ATM has been suggested, utilising MRE11 and RAD51 has also been discussed [40]. MRE11 in isolation was not a prognostic factor, but when combined with RAD51, low expression of both proteins was significantly associated with better response to oxaliplatin, which is part of the Leucovorin, 5-Fluorouracil, and Oxaliplatin (FOLFOX) regimen as first-line treatment for advanced colorectal cancer [40,54]. Furthermore, advanced colorectal cancers with MSI are more sensitive to irinotecan, and low MRE11 or RAD50 expression in these tumours is associated with increased chemosensitivity, and predicts better outcomes [55,56]. This is supported by findings from a cell lineage study that Chk2-deficient cells have defective S-phase checkpoint and hypersensitivity to camptothecins [17]. Since MRN-deficient cells were also deficient in Chk2, this may be useful in predicting treatment response to irinotecan. Given that Leucovorin, 5-Fluorouracil, and Irinotecan (FOLFIRI) is an alternative first-line treatment for advanced colorectal cancer, predicting treatment response based on histological and molecular characteristics of the tumour can be useful to guide treatment choices thereby improving survival and reducing side effects. 

Additionally, early colorectal cancers with MMR-deficiency and high MSI levels have better prognosis and sensitivity to chemotherapy compared to MSS colorectal cancers, but the prognosis remains poor in metastatic disease [57,58]. This suggests the prospect of utilising MRN complex expression as a predictive biomarker for treatment in order to eliminate the burdens of side-effects in patients with a poor baseline and increased chances of treatment resistance. 

## 5. Conclusions 

Multiple studies have sparked discussion of the role of MRE11, RAD50, and NBS1 proteins in multiple cancers. In colorectal cancer, there appears to be an unique relationship between MRN-deficiency and MSI tumours. The positive relationship between MRN expression and colorectal cancer outcomes have not yet been proven in large scale studies, but there is evidence that MRN expression can be useful in selected patient groups. Furthermore, utilising the MRN pathway to improve radiosensitivity is a promising field in colorectal cancer. Substantial additional data will be required to provide standardized assays and interpretation if the MRN is to be used as a biomarker in the clinical setting. Further developments into modulating the MRN pathway will likely provide an answer for targeted therapy, and for improving chemoradiosensitivity for cancer patients. 

## Figures and Tables

**Figure 1 ijms-20-00816-f001:**
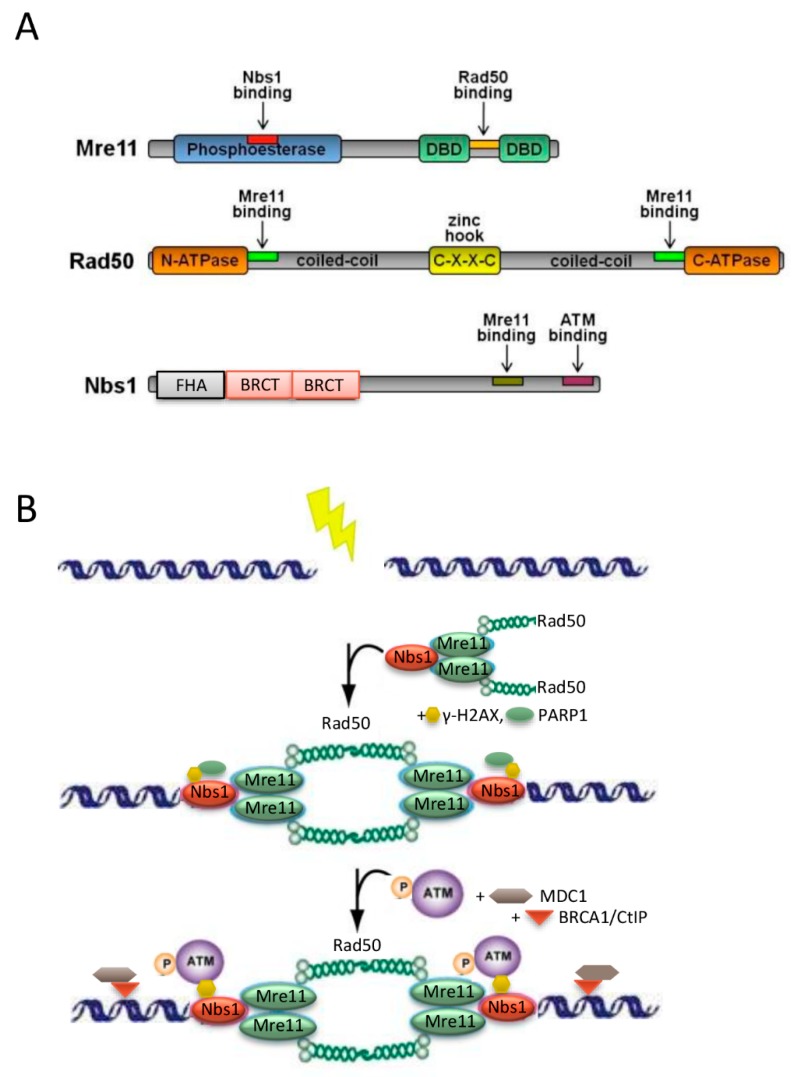
The MRN complex and the DNA-damage response. (**A**) Domain structure of MRE11, RAD50 and NBS1 proteins. (**B**) Model of intermolecular interaction of the MRN complex within the DNA damage response (DDR) pathway, illustrating the initial steps of DDR including exogenous insult (lightning symbol) and the recruitment of MRN complex to the site of DSB by interactions with γ-H2AX (black arrow). See text for details. CXXC, zinc hook; FHA, Forkhead associated domain; BRCT, BRAC1 carboxyl terminus domain; γ-H2AX, the phosphorylated form of H2AX; PARP1, Poly [ADP-ribose] polymerase 1; MDC1, mediator of DNA Damage Checkpoint 1.

**Table 1 ijms-20-00816-t001:** Summary of the correlation between high MRN complex expression and poorer clinicopathological outcomes in colorectal cancer.

Ref.	Tumour Type	Sample Size	Molecule	Results
Alemayehu 2007 [33]	Lynch syndrome (MSI tumours)	28	MRE11 or RAD50	No significant association between high MSI and low MRE11 and/or RAD50 mutations
Ho 2018 [37]	Rectal	265	MRN complex	High MRN expression in TC was associated with higher histological tumour stage, worse DFS, worse OS, and worse DFS and OS in the neoadjuvant radiotherapy subgroup
				Low MRN expression occurred more commonly in the neoadjuvant radiotherapy group
			MRE11	High MRE11 expression was associated with worse OS in the low-grade tumours
			NBS1	High NBS1 expression was associated with worse OS in the high-grade tumours
Chen 2018 [38]	Colorectal	36 CRC cases; Cell lines	RAD50	Higher RAD50 expression was observed in CRC cells and treatment resistant cells compared to non-cancerous mucosa
				Knockout of RAD50 gene sensitises CRC to radiotherapy
Ho 2016 [39]	Rectal	262	MRE11 and ATM combined	High combined MRN and ATM expression in TC was associated with increase LN stage, worse DFS, worse OS and worse DFS in the neoadjuvant radiotherapy subgroup
Ihara 2016 [40]	Colorectal	78	MRE11	Low MRE11 expression was associated with improved oxaliplatin sensitivity and better PFS
				MRE11-negativity had better tumour size reduction compared to MRE11-positivity
			RAD51	RAD51-positivity had poorer progression-free survival compared to RAD51-negativity
Chubb 2016 [41]	Colorectal	1006 CRC cases; 1609 controls	MRE11	MRE11 mutations were found in 3 patients with CRC
Ebi 2007 [42]	Colorectal; Gastric; Lung	2348 controls	NBS1	NBS1 mutation IVS11+2insT was associated with increased risk of gastrointestinal cancers with an odds ratio of 9.43 in colorectal cancer

MSI, microsatellite instable; MRE11, meiotic recombination 11; RAD50, DNA repair protein Rad50; NBS1, Nijmegen breakage syndrome 1; CRC, colorectal cancer; TC, tumour core; DFS, disease-free survival; OS, overall survival; PFS, progression-free survival; ATM, ataxia telangiectasia mutated; LN, lymph node.

**Table 2 ijms-20-00816-t002:** Summary of correlation between high MRN complex expression and favourable clinicopathological outcomes in colorectal cancer.

Ref.	Tumour Type	Sample Size	Molecule	Results
Ho 2017 [9]	Rectal	266	RAD50	Low RAD50 expression in TP was associated with subgroup needing adjuvant treatment
				Low RAD50 expression was associated with worse DFS and OS in low-grade tumour subgroup
				Low RAD50 expression in TC correlated with decrease DFS
Gao 2008 [43]	MSI colorectal; MSS colorectal	208 CRC 171 healthy mucosa 26 LN metastases	MRN complex	High MRN was associated with early tumour stage, MSS status and favourable survival
			MRE11	Higher MRE11 expression was observed in CRC cells compared to normal mucosa; and in MSS tumours, high MRE11 was associated with less local recurrence, high apoptotic activity and favourable survival
			NBS1	High NBS1 was associated with favourable survival in early tumour stage and in MSS tumours

TP, tumour periphery; MSS, microsatellite stable.

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
