# Peer review of "MRN (MRE11-RAD50-NBS1) Complex in Human Cancer and Prognostic Implications in Colorectal Cancer"

_ijms, 2019, doi:10.3390/ijms20040816_

Reviewer 1 Report

This paper summarizes the medical literature that has been done on a relatively interesting and possibly clinically relevant subject, i.e., the role of MRN complex expression in various cancers and their treatment.

The research done is extensive, covering the literature on the subject well. The section explaining the structure and function of the MRN protein complex is nicely done with good figures.  It should be mentioned here that there’s no standardized assays and interpretation of MRN complex currently available for research or clinical use, much less in a defined role as a cancer biomarker.

The rest of the presentation is confusing and roundabout, easily getting readers who are not well acquainted with the research topic lost in all the facts.  In the conclusion, the authors admitted that “there are numerous discrepancies in published findings regarding MRN expression and clinical correlations.” (lines 320-321).

Other observations:

The title is highly misleading because only a portion of the paper was spent on colorectal cancer.

The authors mixed colon cancer topic with other cancers in a puzzling manner, e.g., lines 309-317 under the section titled “…radiosensitivity in colorectal cancer”, a full paragraph on ovarian cancer with low MRI expression responding poorly to radiation therapy was included. Another similar example involved lines 204-205 discussing MRN complex expression in GYN cancers in the section on colorectal cancer.

Table 1 is large and confusing. It should be separated into 2 tables: one showing the studies supporting the association between increased MRN expression and poor cancer outcomes; and the other, showing the studies with opposite findings.

In summary, this paper might contribute to the general knowledge in this field if the authors had decided on a  clear focus for  it. It should either discuss the role of MRN expression in colorectal cancers alone, or include the entire spectrum of MRN expression research in various cancer and cell line studies, reorganize the write-up and change the title.  I think that a smaller, more defined topic would be more enlightening.

Author Response

General: This paper summarizes the medical literature that has been done on a relatively interesting and possibly clinically relevant subject, i.e., the role of MRN complex expression in various cancers and their treatment. The research done is extensive, covering the literature on the subject well. The section explaining the structure and function of the MRN protein complex is nicely done with good figures.  It should be mentioned here that there’s no standardized assays and interpretation of MRN complex currently available for research or clinical use, much less in a defined role as a cancer biomarker. The rest of the presentation is confusing and roundabout, easily getting readers who are not well acquainted with the research topic lost in all the facts.  In the conclusion, the authors admitted that “there are numerous discrepancies in published findings regarding MRN expression and clinical correlations.” (lines 320-321). 

 Response: This has been clarified. 

 Other observations: The title is highly misleading because only a portion of the paper was spent on colorectal cancer. 

 Response: The title of this manuscript has been changed. 

 The authors mixed colon cancer topic with other cancers in a puzzling manner, e.g., lines 309-317 under the section titled “…radiosensitivity in colorectal cancer”, a full paragraph on ovarian cancer with low MRI expression responding poorly to radiation therapy was included. Another similar example involved lines 204-205 discussing MRN complex expression in GYN cancers in the section on colorectal cancer.  

Response: This has been edited and the irrelevant section clarified. 

 Table 1 is large and confusing. It should be separated into 2 tables: one showing the studies supporting the association between increased MRN expression and poor cancer outcomes; and the other, showing the studies with opposite findings.

Response: The original Table 1 has been divided into two tables (Table 1 and Table 2) as suggested, presenting the relationship of protein expression of MRN complex in various cancers with clinicopathological factors accordingly. The focus is now tailored towards studies in colorectal cancer. 

In summary, this paper might contribute to the general knowledge in this field if the authors had decided on a clear focus for it. It should either discuss the role of MRN expression in colorectal cancers alone, or include the entire spectrum of MRN expression research in various cancer and cell line studies, reorganize the write-up and change the title.  I think that a smaller, more defined topic would be more enlightening. 

Response: We appreciate the reviewer’s interest in our work. The title has been changed to cover an intact range of MRN expression studies in human cancer and its prognostic implications in CRC. In addition, the manuscript has been revised with a focus on colorectal cancer.

Reviewer 2 Report

In this review article Situ et. al., discuss about the MRE11-RAD50-NBS1 (MRN) complex. They have reviewed studies showing the functions of MRN complex and its role in cancer progression. The authors further suggest that, MRN complex can potentially be used as a biomarker and putative drug target for colorectal cancer. This review is well written, structured, and covers most of the relevant literatures in the field. It also provides recent developments in this field. The authors should consider taking care of the following minor comments before publishing this article.

Minor comments;

Size of Figure 1 should be increased. Font sizes, particularly of the inserted ones (in the boxes) are too small to read. 

Table 1: Results section overlap between different study groups. Its hard to differentiate the studies from each other. 

Line 157: should be "Interestingly"

Line 158: should be CML cells

Line 165: "leading to" should be put between "thus", and "higher degree". 

Author Response

General: In this review article Situ et. al., discuss about the MRE11-RAD50-NBS1 (MRN) complex. They have reviewed studies showing the functions of MRN complex and its role in cancer progression. The authors further suggest that, MRN complex can potentially be used as a biomarker and putative drug target for colorectal cancer. This review is well written, structured, and covers most of the relevant literatures in the field. It also provides recent developments in this field. The authors should consider taking care of the following minor comments before publishing this article.

Minor comments;

Size of Figure 1 should be increased. Font sizes, particularly of the inserted ones (in the boxes) are too small to read. 

Response: We thank the reviewer for pointing out this.  Figure 1 has been updated with bigger font sizes.

Table 1: Results section overlap between different study groups. It’s hard to differentiate the studies from each other. 

Response: Table 1 has been divided into two tables, presenting the relationship of protein expression of MRN complex in colorectal cancer with clinicopathological factors accordingly.

Line 157: should be "Interestingly"

Response: Corrected

Line 158: should be CML cells

Response: Corrected

Line 165: "leading to" should be put between "thus", and "higher degree". 

Response: Corrected

Round  2

Reviewer 1 Report

The revised paper with its enhanced figures, tables and focused discussion is much better, providing great insight into a complex subject.

I would suggest one minor change: 

Line 232  MRN protein complex and sensitivity to chemoradiation in colorectal cancer

This section does discuss cancer response to both radiation and chemotherapy in relation to the MRN complex.

Author Response

Thank you for the comment in regards to Line 232. This has now been amended.